# Entropy-based discrimination between translated Chinese and original Chinese using data mining techniques

**Kanglong Liu** [1]*, **Rongguang Ye** [2], **Liu Zhongzhu** [3], **Rongye Ye** [3]

**1** Department of Chinese and Bilingual Studies, The Hong Kong Polytechnic University, Hong Kong, China, **2** School of Applied Mathematics, Guangdong University of Technology, Guangdong, China, **3** Department of Mathematics and Statistics, Huizhou University, Guangdong, China

* klliu@polyu.edu.hk

**Data Availability Statement:** All relevant data in this study are available at https://osf.io/asdbh/.

**Funding:** The author(s) received no specific funding for this work.

## Abstract

The present research reports on the use of data mining techniques for differentiating between translated and non-translated original Chinese based on monolingual comparable corpora. We operationalized seven entropy-based metrics including character, wordform unigram, wordform bigram and wordform trigram, POS (Part-of-speech) unigram, POS bigram and POS trigram entropy from two balanced Chinese comparable corpora (translated vs non-translated) for data mining and analysis. We then applied four data mining techniques including Support Vector Machines (SVMs), Linear discriminant analysis (LDA), Random Forest (RF) and Multilayer Perceptron (MLP) to distinguish translated Chinese from original Chinese based on these seven features. Our results show that SVMs is the most robust and effective classifier, yielding an AUC of 90.5% and an accuracy rate of 84.3%. Our results have affirmed the hypothesis that translational language is categorically different from original language. Our research demonstrates that combining information-theoretic indicator of Shannon's entropy together with machine learning techniques can provide a novel approach for studying translation as a unique communicative activity. This study has yielded new insights for corpus-based studies on the translationese phenomenon in the field of translation studies.

## Introduction

Translation plays an important role in this age of globalization and increased cross-cultural communication and has therefore received increasing attention from researchers working in various fields [1, 2]. As a by-product of cultural fusion and communication, translation is under the influence of both source and target languages. It has often been assumed that translation is a unique language which is heterogeneous to both the source and target language. Translation is in nature a type of mediated language which "has distinctive features that make it perceptibly different from comparable target language" [3]. Such an assumption has led translation to be seen as "derivative and of secondary quality and importance" [4] and "rarely considered a form of literary scholarship" [5]. Viewed as a substandard language variety,

**Competing interests:** The authors have declared that no competing interests exist.

translation has also been labelled as the "third code" [6] situated between source and target languages and "translationese" [7].

In the field of translation studies, scholars have been intrigued in identifying the uniqueness of translational language that separates it from native writing. This line of research has largely been conducted using corpus-based approaches spearheaded by Baker [8, 9]. As a pioneering researcher, Baker [8] specifically pointed out that translated texts need not be compared vis-à-vis their source texts. Instead, scholars can examine the peculiarities of translational language using a comparable corpus approach, i.e., comparing a corpus of translated texts with one of native writing of the same language which is comparable in genre and style. Instead of comparing source text with translation, Baker contended [9] that scholars need to effect a shift in the focus of theoretical research in the field of translation studies by "exploring how text produced in relative freedom from an individual script in another language differs from text produced under the normal conditions which pertain in translation, where a fully developed and coherent text exists in language A and requires recoding in language B". The use of comparable corpora can fulfil such a purpose. Since Baker's proposal, this field of research has quickly gained momentum and is widely referred to as Corpus-Based Translation Studies (CBTS). Specifically, Baker [8] proposed the concept of translation universals claiming that translational language is ontologically different from non-translational target language due to the translation process irrespective of the influence from either language systems. The research agenda proposed by Baker has continued to captivate the interest of CBTS researchers. A vast number of studies have been devoted to the investigation of the unique features of translational language including "simplification (translation tends to simplify language use compared to native writing)" [10], "explicitation (translation tends to spell out the information in a more explicit form than the native writing" [11, 12], "normalization or conservatism (translation tends to conform to linguistic characteristics typical of the target language" [13], and "levelling out (translation tends to be more homogeneous than native texts)" [14]. Although these efforts have yielded some new insights as to the nature of translational language, most of these studies have confined their research to the study of manually-selected language features. On the other hand, in the neighbouring field of computational linguistics, researchers have successfully used machine learning techniques to conduct text classification tasks. Clearly, as an interdisciplinary field of study, translation studies needs to look across the disciplinary fence towards computational linguistics for rejuvenating corpus-based investigations of translational language.

Based on two balanced comparable corpora, the current study makes use of text classification techniques to distinguish translated texts from non-translated original writings of the same language (i.e., Chinese in this case). We first calculated Shannon's entropy of seven language features including character, wordform unigram, wordform bigram, wordform trigram, POS unigram, POS bigram and POS trigram from two balanced comparable corpora (translated vs non-translated), and applied four data mining techniques including Support Vector Machines (SVMs), Linear discriminant analysis (LDA), Random Forest (RF) and Multilayer Perceptron (MLP) to distinguish translated Chinese from non-translated Chinese based on these seven features.

Instead of focusing on isolated language features to identify the uniqueness of translational language, our study has integrated Shannon's information theory and data mining techniques to study translational language. It is demonstrated that the use of informational-theoretic indicator of entropy together with automated machine learning can be an innovative approach for the classification task. We believe that the research can be of interest to CBTS researchers from a methodological point of view. In our study, we have also shown that the machine learning methods have been more robust than statistical significance analysis methods. The following

part of the article is structured as follows: Section 2 provides a brief review of related work on the study of translationese and text classification. Section 3 introduces the two corpora, based on which we calculated seven entropy-based features. Section 4 presents the different machine learning techniques that we used in this study, and the results are reported in Section 5. The discussion of results and conclusion are given in Section 6 and 7 respectively.

## Related work

### Previous studies on translation features

Scholars have long noticed that translation carries language features that are different from the original writing of the target language [7]. Translation is thus conceived as an unrepresentative variant of the target language system [15]. Newmark [16] believed that translationese is a result from the translators' inexperience and unawareness of the interference from the source language. However, not many scholars hold a derogatory view towards translationese. For example, Baker [8] pointed out that any translation output is inevitably characterized with unique language features that are different from both the source and target languages. In fact, the quest into translation universals (TU) proposed by Baker [8, 9] has sparked a new wave of corpus-based research into the unique features of translational language and greatly enhanced the status of translation studies as an independent field of research. With its capacity of handling large amount of data, corpus has been used to explore various translation topics and increasingly accepted by translation researchers. CBTS research has helped foster a transition from an overreliance on source texts to a systematic investigation of how translation plays out in the target language system [17].

Following Baker's proposal, a plethora of studies investigating translationese specifically adopted a monolingual comparable corpus consisting of translated texts and non-translated original texts of the same language. Such studies are often exploratory in that the frequencies of manually selected features in both translated and non-translated corpora are calculated to show if significant differences exist between the two types of texts. Over the years, researchers have found that translated texts tend to demonstrate a lexically and syntactically simpler [17, 18], more explicit [12, 19] and more conservative [20] trend than comparable non-translated texts of the same language. In addition to these features, translated texts have been found to underrepresent target language specific elements which do not have equivalents in the source language [21, 22] and carry source text language features due to the source language shinning-through influence [23].

In the past two decades, TU research has greatly advanced the development of translation studies despite various controversies. Overall, research on TU has to a large extent been confined to Indo-European target languages such as English [13, 18], Finnish [24], Italian [20], Spanish [25], German [26] and French [11]. As has been argued by Xiao and Dai [3], it is important to look into typologically distant languages such as English and Chinese to investigate the translationese phenomenon as features derived from such a language pair might be distinctively different from closely-related languages. Nevertheless, these TU studies in the past two decades have been fruitful in identifying some disproportional representation of lexical, syntactic and stylistic features in translated and non-translated texts. The results have supported the claim that translation is distinct from non-translation in a number of linguistic features. As research advances, notably with the maturity of corpus tools and technology, researchers have begun to examine translational features in other languages than English. In the Chinese context, Chen [27] found that translated Chinese overused connectives than non-translated Chinese based on a parallel corpus of English-Chinese translation in the genre of popular science writing. Xiao and Yue's study [28] found that translated Chinese fiction

contains a significantly greater mean sentence length than native Chinese fiction, which contradicts the previous assumption [29] that the general tendency in translation is to adjust the source text punctuations in order to conform to the target language norms, thus resulting in similar sentence lengths between the source texts and comparable target texts. In a more sophisticated study based on a corpus of translated Chinese and a comparable corpus of native Chinese, Xiao [30] found that translated Chinese has a significantly lower lexical density and uses conjunctions and passives more frequently than native Chinese, which corroborates the simplification, explicitation and source language shinning-through hypotheses respectively. It should be noted that the researcher has warned that some language features might be genre-sensitive instead of translation-inherent (e.g., sentence length). It is worth noting that machine learning algorithms have successfully been applied to the discrimination of different genres [31, 32], indicating that genre can be an important variable in this regard. Research on translationese should also consider genre variation by carefully selecting the right text types [30]. In a follow-up study, Xiao [12] also found that translated Chinese tends to use fixed and semi-fixed recurring patterns than non-translated Chinese, presumably for the sake of improving fluency. The researcher suspected that the higher frequency of word clusters in translated Chinese is a result of interference from the English source language which is believed to contain more word clusters than the native Chinese language. Again, this can be attributed to the source language shining through influence. Building on Xiao's investigations of translated Chinese features [12, 30], Xiao and Dai [3] further found that translated Chinese differs from native Chinese in various lexical and grammatical properties including the high-frequency words and low-frequency words, mean word length, keywords, distribution of word classes, as well as mean sentence segment length and several types of constructions. Their research further corroborates that translated Chinese has possessed some uniqueness as "a mediated communicative event" [8]. Using collocability and delexicalization as operators, Feng, Crezee and Grant [33] verified and confirmed the simplification and explicitation hypotheses in a comparable Chinese-to-English corpus of business texts. They found that translated texts are characterized with more free language combinations and less bound collocations and idioms.

Based on the forgoing review on translated Chinese, we can see that researchers tend to adopt a bottom-up exploratory approach by comparing translated texts against non-translated texts using isolated features. Such a research design has some inherent limitations. For example, the examination of translationese or translation universals is often confined to manually-selected linguistic indicators. The use of such indicators runs the risks of "cherry picking" by deliberately selecting those indicators to confirm the researcher's perceptions or hypothesis [17]. There is clearly a lack of global and holistic features in this line of research. Apart from the selection of holistic language features, it should be seen that resorting only to descriptive statistics to compare two sets of corpora is limited as the differences observed might not be statistically significant. We contend that information-theoretic indicators can be operationalized to probe into the categorical differences between translated and non-translated original texts.

To address such an issue, more researchers have begun to turn to the neighbouring field of computational linguistics to study translationese. For example, Fan and Jiang [34] used mean dependency distances (MDD) and dependency direction as indicators to examine the differences between translated and non-translated English. Their results showed that translated English uses longer MDD and more head-initial structures than non-translated English. Besides, starting from Baroni and Bernardini [35], more researchers have turned to the use of machine learning techniques to classify translation from original writing. Such an approach has greatly overcome the subjectivity resulted from cherry-picked features. Their work, though preliminary in terms of the use of isolated language features and specific text types (i.e., collection of articles from one single Italian geopolitics journal) and single machine learning model

(i.e., use of Support Vector Machines or SVMs), represents one of the pioneering studies to utilize machine learning to the investigation of translational language. In the next section, we will review some relevant studies using machine learning.

## Machine learning and classification of translational language

Based on a monolingual Italian corpus consisting of original and comparable translated articles of a geopolitics journal, Baroni and Bernardini [35] made use of machine learning to classify translated texts from non-translated ones using n-grams of different unit types (i.e., wordform, lemma, POS tags). They found that an ensemble of SVMs (Support Vector Machines) reaches 86.7% accuracy with 89.3% precision and 83.3% recall on the classification task, demonstrating that the translationese hypothesis can be verified through machine learning techniques. Their study further showed that SVMs-based algorithms performed much better than experienced translation practitioners on the classification task. Following the inspiring work of Baroni and Bernardini [35], Kurokawa et al. [36] used a mixed text feature by replacing content words with their corresponding POS tags while retaining the function words on a corpus of Canadian Hansard. By performing classification at both the document and the sentence level, they found that translation models trained on English-translated-to-French parallel texts performed much better than the ones trained on French-translated-to-English, when the statistical machine translation (SMT) task was based on English-French translation. Their research findings were further corroborated by latter studies [37, 38] that found translation direction serves as an important variable for SMT and proposed to adapt translation models in the investigation of the features of translational language.

All the studies mentioned above have shown that it is possible to characterize translational language using machine learning techniques. Though all these studies were not aimed at confirming the existence of certain TU candidates, almost all these studies using machine learning algorithms have made use of the language features previously identified by translation scholars [10, 18, 39] in performing the classification. In this line of research, the study of Ilisei et al. [40] represents one of the innovative studies utilizing machine learning methods to test and prove the simplification hypothesis. They trained the classifier on POS unigrams together with the preselected "simplification features" and then evaluated the success rate with different combinations of features. Their study has found that lexical richness, together with sentence length and the proportion of function words to content words, are the top three effective features for characterizing translated Spanish from non-translated Spanish. The accuracy rate of 97.62% shows that the simplification hypothesis is corroborated in translated Spanish (note that all these three features are considered representative of the simplification hypothesis). Such a methodology was further replicated in translated Romanian to confirm the simplification hypothesis [41] and translated Spanish and translated Romanian to confirm the explicitation hypothesis [42]. In the same vein, Volansky et al. [43] also investigated the features of translationese using supervised machine learning (i.e., SVMs classifiers) with ten-fold cross-validation evaluation and found that some language features serve as robust indicators to confirm the translation universals hypotheses while some language features perform less than satisfactorily. The use of machine learning techniques to attest the translation universals have yielded some new insights into the translationese phenomenon both methodologically and empirically.

However, it should be noted that most of these studies are confined to texts from a homogenous corpus or closely-related language pairs. The accuracy rate decreases considerably when the classifiers are applied to texts of different genres or translations with a different source language. Such issues were addressed in Koppel and Ordan's study [44] that applied machine

learning techniques on the Europarl Corpus which contains translated English from five source languages. The accuracy rate of the classifiers was close to 100% when evaluated on the language pair that the classifiers were trained on. However, the accuracy rate declined substantially when evaluated on translated English with a different source language. The same situation occurred when the classifier was tested on a different genre other than the original training corpus. The expanded version of Europarl Corpus was further used by Volansky et al. [43] to study simplification, explicitation, normalization, and interference, which are the four main translation universals frequently investigated by translation scholars. By operationalizing different features of simplification, they found that TTR achieved slightly higher than 70% accuracy, lexical density about 53% and sentence length only 65%. Based on the classification results, the researchers concluded that the simplification hypothesis was rejected since translations from seven source languages including Italian, Portuguese and Spanish have longer sentences than the non-translated English. The research further found that translations tend to carry the affix features from the source languages, which corroborates the interference hypothesis.

Overall, it can be seen from the foregoing review that translated texts are categorically different from original ones and machine learning classification algorithms are effective in differentiating them with a rather high accuracy. Earlier works by CBTS scholars in the quest of translationese features have laid out the groundwork for machine learning-based classification research of translationese which has gradually developed into a prolific field of inquiry. As research advances, we have also seen more studies using data mining techniques [45, 46] which have advanced our understanding of the translationese phenomenon and translation as a unique variant of language communication. However, it should be noted that like CBTS studies on translationese features, the machine learning-based studies in this line of enquiry are also predominantly based on European languages. There is clearly a lack of research on Chinese or Asian languages in general. Of the few studies, Hu et al. [47] is one of the pioneering studies to characterize translated Chinese using machine learning methods. Based on a comparable corpus of translated and original Chinese, they showed that using constituent parse trees and dependency triples as features can attain a high accuracy rate in classifying translated Chinese from original Chinese. In a follow-up study on translated Chinese, Hu and Kübler [48] operationalized a number of language-related metrics related to different TU candidates including simplification, explicitation, normalization and interference and tested them with machine learning algorithms. Their research shows that translations differ from non-translations in various features, confirming the existence of some translation universals. Specifically, the interference-related features achieved a very high accuracy, indicating that translations are under the influence of the originals. This point can be attested by their research findings that translations from Indo-European languages share more similarities while those translated from Asian languages of Korean and Japanese are more similar to each other. In the next, we will address the limitations of this research area and present a novel method integrating data mining techniques with entropy-based measures to differentiate translationese from original writing.

## Methodology

### Research questions

As has been seen in the foregoing review of existing literature, the research using machine learning algorithms to study translationese has attracted the attention from various fields of researchers. However, this line of research has some methodological limitations. First, the use of machine learning algorithms is not well justified in most studies and is often confined to the

use of SVMs classifier [47, 48]. From a machine learning perspective, it is important to also try other classifiers to assess the classification performance. Second, research in this area predominantly focuses on confirming the existence of translationese with the use of limited language properties cherry-picked by the researchers. Almost all the studies based their investigations on the features identified by CBTS researchers, for instance, simplification features such as type token ratio, mean sentence length from Laviosa [18], explicitation features such as the use of cohesive markers from Chen [27]. Notwithstanding its convenience in supplying researchers with an ensemble of language features or metrics, the use of cherry-picked features to confirm the existence of TU can be methodologically flawed, which has long been under criticism by mainstream translation theorists [26]. Thirdly, a large number of studies are limited to the use of single-genre corpus, e.g., the news genre from one single source [48]. In the field of translation studies, researchers have warned that one major weakness of TU research is that the assumed translationese features are considered to be independent of genre or language pair which can actually make a big impact on the profiling of translational language [19, 30]. In view of these limitations, we believe there is clearly a need for adopting holistic information-theoretic features to study translations based on a multi-genre corpus. The current study uses entropy to examine if translation differs from non-translation from an information-theoretic perspective. It is believed that entropy, which has proved fruitful in numerous research areas including informational technology, computational linguistics and biochemistry, can overcome the limitations of cherry-picked features and shed light on translationese research.

The purpose of the study is to explore whether translated Chinese is categorically different from non-translated original Chinese based on a multi-genre corpus using entropy-based indicators. To achieve such a purpose, we will test a number of machine learning algorithms. The following two research questions will be addressed:

1. Can machine learning techniques distinguish translated Chinese from non-translated Chinese using entropy-based features?

2. If the answer to the first question is yes, then what are the top-performing entropy-based features for the classification?

As an information-theoretic indicator, entropy has been successfully used to measure information richness and text complexity [49, 50]. For this reason, entropy is directly relevant to a number of translation universals including simplification and explicitation mentioned earlier. In this study, we operationalized seven entropy-based indicators, including (1) character entropy, (2) wordform unigram, (3) wordform bigram, (4) wordform trigram, (5) POS (Part of Speech) unigram, (6) POS bigram, and (7) POS trigram. Wordforms are the morphological realizations of grammar and the entropy of wordforms can help predict the average vocabulary richness of a text. However, the wordform entropy cannot measure the syntactic complexity of a text. For such a reason, researchers have instead turned to the use of POS forms as an indicator of syntactic complexity as POS has "attained a certain degree of abstraction for words" [17, 27]. Note that this can be linked to the simplification universal that translated texts tend to be simpler than non-translated texts [51, 52]. For this reason, it is deemed viable to use entropy-based indicators to distinguish translated texts from non-translated ones. The study is based on two comparable corpora, i.e., translated Chinese and non-translated Chinese. Details of the two corpora are provided in the subsection below.

## Corpora and data processing

We used two comparable corpora in this study, i.e., The ZJU Corpus of Translational Chinese (ZCTC) which consists of Chinese texts translated from English and The Lancaster Corpus of

**Table 1. Genres and Text types in LCMC and ZCTC.**

| Genres | Text type | Samples | Proportion |
|---|---|---|---|
| Press | Press reportage | 44 | 8.8% |
| | Press editorial | 27 | 5.4% |
| | Press reviews | 17 | 3.4% |
| General Prose | Religious writing | 17 | 3.4% |
| | Instructional Writing | 38 | 7.6% |
| | Popular lore | 44 | 8.8% |
| | Biographies and essays | 77 | 15.4% |
| | Reports and official documents | 30 | 6% |
| Academic | Academic prose | 80 | 16% |
| Fiction | General fiction | 29 | 5.8% |
| | Mystery and detective fiction | 24 | 4.8% |
| | Science fiction | 6 | 1.2% |
| | Adventure fiction | 29 | 5.8% |
| | Romantic fiction | 29 | 5.8% |
| | Humor | 9 | 1.8% |
| | Total | 500 | 100% |

Mandarin Chinese or LCMC consisting of original Chinese. The compilers deliberately designed the corpora based on the structure of The Freiburg-LOB (FLOB) Corpus. Each corpus has one million words sampled from four major genres in order to objectively reflect natural language use. The two corpora can be considered as the translated and original Chinese counterparts of FLOB as they are comparable in sampling methods and period [53]. The four macro genres of LCMC and ZCTC are news, general prose, academic prose, and fiction (see Table 1). It is believed that the use of multi-genre balanced corpora can better represent translation and non-translation in the Chinese language in this study. The two corpora LCMC and ZCTC have been fruitfully studied to generate some new insights regarding the translationese phenomenon in Chinese [54].

## Segmentation and data processing

Text segmentation and annotation is a necessary step in corpus linguistics and corpus-based translation research, notably in the Chinese language. As a highly contextual language which has no clear delimitation between words [55], Chinese has reported a lower accuracy rate in segmentation and annotation compared to other languages. In this study, we used The Stanford CoreNLP, which comprises The Stanford Parser [56] and the Stanford Chinese POS Tagger [57], to conduct the segmentation and annotation. The software reports an overall accuracy of 93.65% and has been applied to a number of corpus-based studies in Chinese. Previous research has identified that entropy value is subject to text length [58]. In order to ensure that the text length is consistent across both corpora, we trimmed each corpus to 1500 Chinese words per text to ensure consistency upon noticing that the original texts in the two corpora are not comparable in length. After the processing, we were able to retain the same number of files as per the original corpus design. In this study, all the punctuations were redacted to ensure that the punctuation use does not affect the entropy values calculated on each text. We then calculated seven entropy-based measures based on Formula (1).

$$S(p) = -\sum_{x \in X} p(x)\log_2 p(x) \tag{1}$$

The entropy-based calculation has proved more advantageous than the traditional methods such as type-token ratio (TTR) because it takes both frequencies and distribution of tokens into consideration. Generally speaking, a larger entropy value can help predict that the text contains more unique wordforms (or POS forms), and these words (or POS forms) are more evenly distributed in the text.

## Machine learning models

In this study, we used four machine learning classifiers, i.e., SVMs, LDA, RF, and MLP, to build the classification models, and then we compared the AUC value and accuracy rate of the four classifiers to identify the optimal one. We selected the two top-performed classifiers and ranked the feature importance based on AUC values. Through comparing the output feature importance of the two top-performed classifiers, we were able to identify the features that have excellent predictability in differentiating translation from non-translation. The feature data were then visualized in density scatter plots based on SHAP values to show their predictive ability and contribution in the differentiation task. Note that SHAP (SHapley Additive exPlanations) is a game theoretic framework to assess the output of machine learning models [59]. Similar to Shapley values in game theory which compute the contribution that each player brings to the game, SHAP values compute the contribution that each feature brings to the prediction made by the model. Because SHAP values are in nature additive, individual (local) SHAP values can be aggregated for global explanations. SHAP can be used as a foundation for machine learning analysis such as model monitoring, fairness and cohort analysis in addition to accuracy rate. In the next, we briefly introduce the four classifiers used in the current study.

### Support vector machines (SVMs)

SVMs work by maximizing the distance between borderline cases and a decision boundary. Like logistic regression, SVMs have also been used in a wide range of scientific fields. SVMs were originally developed in the 1990s and have since been used in supervised learning in various areas as diverse as detecting malware and spam [60], financial fraud [61], and classifying neurological scans [62]. Prior to the new deep learning era (2012), SVMs were a mainstay for computer vision tasks such as facial recognition [63] and image classification [64]. SVMs have also been used in NLP tasks, such as document classification, named entity recognition, sentiment analysis, and argument mining. A few attempts have even been made to use SVMs to distinguish between translated and native-language texts [35, 45, 48]. Therefore, SVMs is also adopted in this study on merit of its successful performance in various classification studies.

### Linear discriminant analysis (LDA)

Linear discriminant analysis (LDA) seeks an optimal linear transformation by which the original data is transformed to a much lower dimensional space, with the goal of finding a linear transformation that maximizes class separability in the reduced dimensional space [65]. As one of the most important supervised linear dimensional reduction techniques, LDA seeks to use low-dimensional representation from the original high-dimensional data through a transformation formula by maximizing the between-class scatter and minimizing the within-class scatter. LDA has been successfully used in various applications including face recognition [66], speech recognition [67] and text classification [65, 68].

### Random forests (RF)

Decision trees are a supervised classification technique that chooses features on which to classify data, with the goal of finding the features that best predict outcomes. Random forests work by generating many decision trees from random samples of the data and selecting the features which reliably best predict the outcomes. Random forests have, like the other classification algorithms, been used across a wide range of fields and for many very distinct tasks, from predicting how sensitive tumors are to drugs [69], to predicting the onset of civil wars [70], to predicting the role of nuclear proliferation on interstate conflict [71].

Again, as with the other classifiers, random forests have been applied to prediction and classification problems in natural language. A few of the more novel applications are sarcasm detection [72], predicting the severity of mental symptoms [73], and argument mining on Twitter [74].

### Multilayer perceptron (MLP)

MLP is a neural network model inspired by the biological nervous system that works as global approximators to implement any given nonlinear input-output mapping. MLP makes use of a supervised learning technique (i.e. backpropagation) for machine learning. MLPs performs computational tasks by processing neurons called perceptions. The neurons are grouped in multiple layers that are connected through weights [75]. In the field of machine learning, classifiers based on artificial neural network have proved to be effective tools in classification tasks. As one of the artificial neural network classifiers, Multilayer perceptron (MLP) has successfully been applied in many applications including pattern recognition [76], signal processing [77] and text classification [78] on merit of its fast convergence and easy implementation. This study uses the sigmoid functions as the activation functions of the MLP model.

While these classifiers have been applied to NLP (Natural Language Processing) tasks, they have not been widely applied to the task of distinguishing translational from native language. Only a few papers have used SVMs (see Section 3 for a detailed review) to distinguish between translations and native language texts, and the algorithms which have proved successful in related classification tasks are yet to be applied in this area. Our application of these machine learning classifiers, therefore, offers a novel contribution to this line of enquiry.

## Results

The dataset includes 500 original texts (i.e., LCMC) and 500 translated texts (i.e., ZCTC) with a total of 1000 samples. The data were randomly divided into a training set (70% of the data) and a test set (30% of the data). In order to distinguish between original texts and translated texts, this study trained four classifiers (i.e., SVMs, LDA, RF, and MLP) in the training set for classification. The four classifiers were implemented in Python. After training, we evaluated the performance of these classifiers in the test set with two evaluation indicators (i.e., AUC and Accuracy). Table 2 shows the results of the four classifiers. We observed that SVMs (linear) has an AUC of 90.49% and an accuracy rate of 84.33%, which performed better than other

**Table 2. Performance evaluation of the four classifiers on the test set.**

| Classifier | AUC (%) | Accuracy (%) |
|---|---|---|
| **SVMs (linear)** | **90.49** | **84.33** |
| LDA | 90.03 | 81.33 |
| MLP | 89.97 | 83.33 |
| RF | 88.15 | 82.00 |

**Table 3. Importance coefficient and ranking of features in SVMs model.**

| Feature | Coef | Important Rank |
|---|---|---|
| POS bigram | -7.9062 | 1 |
| POS trigram | 5.4895 | 2 |
| Wordform trigram | -5.3873 | 3 |
| Wordform unigram | 3.5080 | 4 |
| Wordform bigram | 3.0918 | 5 |
| POS unigram | -2.9562 | 6 |
| Character | -2.5134 | 7 |

classifiers (shown in bold numbers in Table 2). We also observed that LDA ranks second with an AUC of 90.03%.

In our study, the size of the feature importance coefficient is used as the index to evaluate the importance of the feature. The feature with a positive feature importance coefficient is more likely for the classifier to predict as original language (i.e., LCMC), and the feature with a negative feature importance coefficient is more likely for the classifier to predict as translated language (i.e., ZCTC). The larger the absolute value of the feature importance coefficient, the more important the feature is to the classification. In view of the highest AUC value obtained from SVMs and LDA, we show the feature importance coefficient and ranking of SVMs model and LDA model in Tables 3 and 4. It can be seen from Table 3 that POS trigram is an important feature for the classifiers to predict as original text, and POS bigram is an important feature for the classifier to predict as translated text. Table 3 shows that POS bigram is the most important feature with an importance coefficient of -7.962, the second-ranked POS trigram coefficient is 5.4895, and the third-ranked wordform trigram coefficient is -5.3873. In addition, it can be seen from Table 4 that POS bigram is the most important feature with a coefficient of -22.7247, the second-ranked POS trigram coefficient is 16.4629, and the coefficient of the third-ranked wordform trigram is -12.2450. Clearly, the ranking of the top three features remain relatively the same across SVMs and LDA classifiers.

SHAP values can be effectively used to explain the decision-making process of various machine learning models [59]. As mentioned earlier, SHAP has the advantage of reflecting the positive and negative effects of each sample. Based on the results of the best performing classifier SVMs in this study, we used the SHAP values to illustrate the various features of the SVMs model. In Fig 1, each row represents a feature, the colours from blue to red represent the entropy values of the feature, with blue representing the low and red the high values. We can tell from Fig 1 that smaller SHAP value leads to a higher probability of predicting the text to be a translated one. We can observe that the higher POS bigram value corresponds to a smaller SHAP value, indicating that it is more likely for a text with higher POS bigram entropy value

**Table 4. Importance coefficient and ranking of features in LDA model.**

| Feature | Coef | Important Rank |
|---|---|---|
| POS bigram | -22.7247 | 1 |
| POS trigram | 16.4629 | 2 |
| Wordform trigram | -12.2450 | 3 |
| Wordform bigram | 7.5781 | 4 |
| Wordform unigram | 4.3513 | 5 |
| Character | -3.6422 | 6 |
| POS unigram | -0.4390 | 7 |

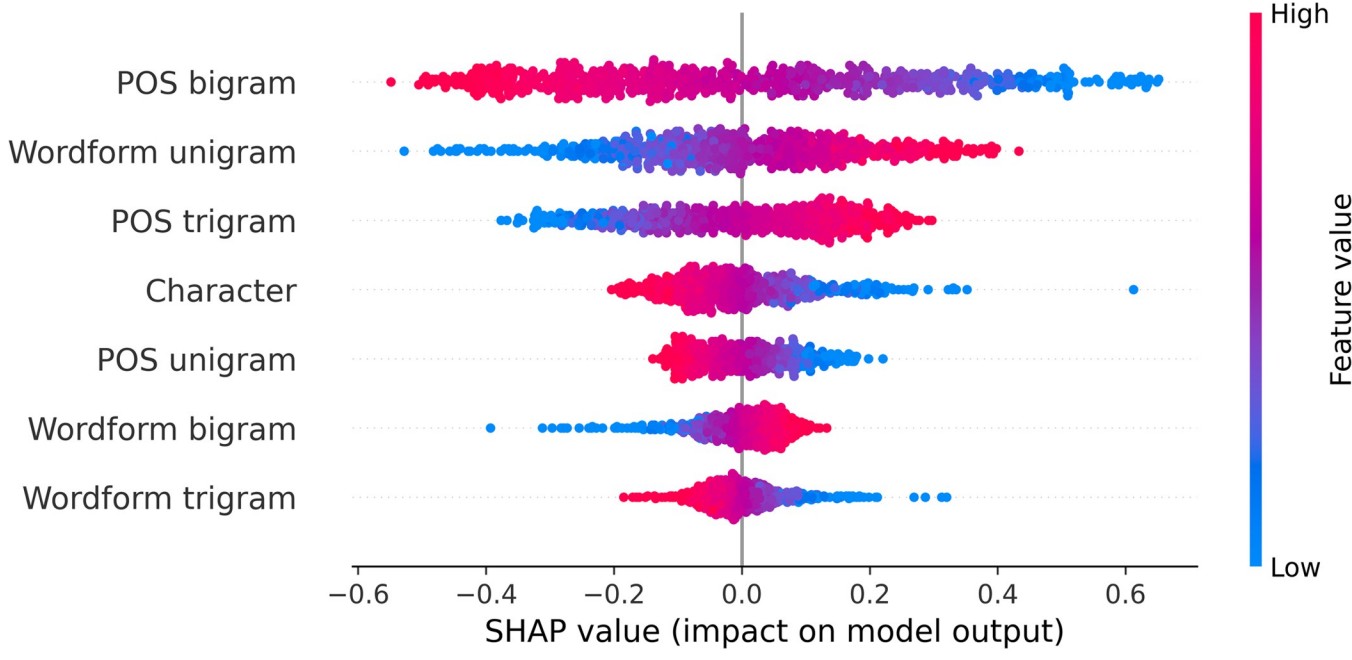

**Fig 1. SHAP value in the SVMs model.**

to be a translated one. On the other hand, a higher wordform unigram entropy value is correspondent to a higher SHAP value, meaning that the text is more likely to be an original one.

In order to study how the between-feature interaction affects the performance of the classifier, we used a partial dependence plot to explore the interactive effect. Based on the top three features of SVMs and LDA (POS bigram, POS trigram, Wordform trigram), we created contour plots depicting effects of interaction between every two features in the SVMs model (Fig 2). When the interaction between POS bigram and POS trigram is examined while other features remain constant, it can be observed that both have a certain degree of influence on the final classification result (see Fig 2(A)). When POS bigram is small (e.g., POS bigram< 6.15), the change of POS trigram value has little effect on the final classification result, and it is more likely for the classifier to predict the text to be an original one. When POS trigram is small (e.g., POS trigram< 7.96), POS bigram has little influence on the final classification result, and it is more likely for the classifier to judge the text to be a translated one. When we examine the interaction between POS bigram and POS trigram while other features remain constant, it can be observed that both have a certain degree of influence on the final classification result (see Fig 2(B)). When POS bigram is small (e.g., POS bigram< 6.15), we can see that the change of POS trigram has little effect on the final classification result, and it is more likely for the classifier to predict the text to be an original one. When POS trigram is small (e.g., POS trigram< 7.96), we can see that POS bigram has little influence on the final classification result, and it is more likely for the classifier to judge the text to be a translated one. Finally, in Fig 2(C), when the POS trigram is about 8.8 and the Wordform trigram is about 10, it is most likely for the classifier to predict the text to be original.

## Discussion

In this paper, we have proposed using machine learning methods to classify translated Chinese from non-translated Chinese. We have demonstrated that the differences between translation

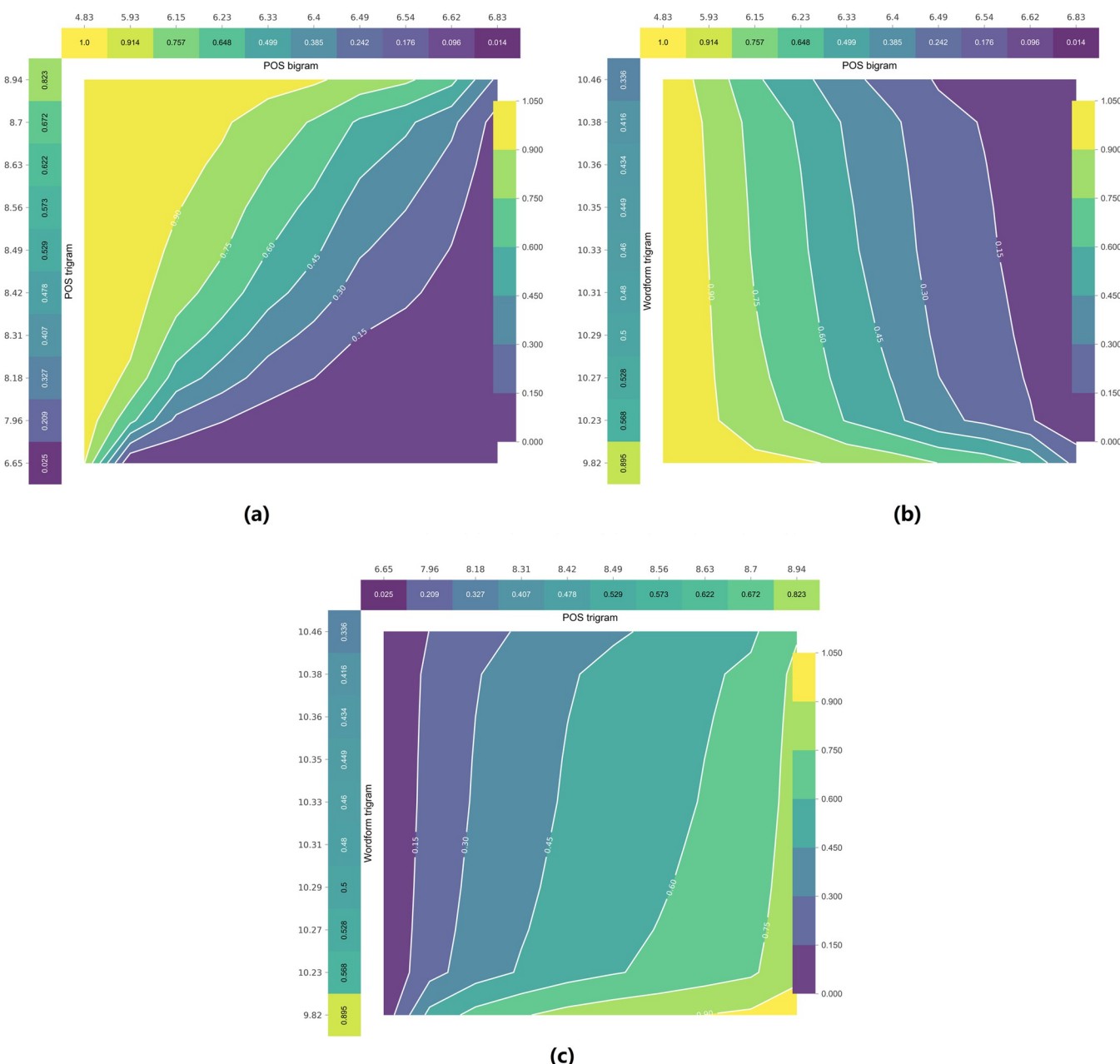

**Fig 2. Two-feature Interaction effect in contour plot.** (where (a) is the interaction between POS bigram POS trigram, (b) is the interaction between POS bigram and Wordform trigram, (c) is the interaction between POS trigram and Wordform trigram).

and non-translation can be learned by machine learning algorithms irrespective of the genres using entropy-based features. The seven entropy-based features we extracted include character, wordform and POS entropies which were computed from two multi-genre balanced corpora (translated and non-translated). Four machine learning classifiers, i.e., SVMs, LDA, RF and MLP, were utilized to conduct the classification tasks based on the features. Our research shows that the translationese phenomenon in Chinese can be approached from an information-theoretic perspective using machine learning methods.

Different from previous research which was based on specific language features, our research has shown that using information-theoretic indicator of entropy can also achieve a

very high success rate in classification. Our study of using entropy-based features was mainly inspired by the assumptions that translation tends to be simpler and more explicit than non-translation [8, 9]. We believe that the simplification and explicitation assumptions can be extrapolated from entropy-based measures which calculate information richness. As entropy is a measure of randomness and chaos [79], the information contained in a text can be measured for its regularity and certainty. Therefore, we can interpret the information regularity of a text as basically the negative of its entropy. That is, the more probable the text message, the less information it carries. Based on our research, we found that translation differs from non-translation in lexical and syntactic complexity as measured by the wordform and POS entropy. In sum, it is believed that the use of entropy can be a promising avenue for corpus-based translationese research and our study has shown that machine learning methods can distinguish the translated Chinese from original Chinese with a rather high accuracy.

As has been noted in previous machine learning-based research, apart from the influence of language-pairs, translationese can also be subject to the variables of genres and registers [43]. Although this issue has long been noted by translation researchers [12], they seemed slow in using balanced corpora to address such an issue. Previous studies in this line of inquiry were often confined to one single genre such as geopolitical texts [35], news and the proceedings of the European Parliament [43]. Being vigilant of the limitations, Volansky et al. [43] have called for similar research to be done in a well-balanced comparable corpus based on typologically distant languages. Our research echoes their call by using two comparable multi-genre balanced corpora to examine translationese in Chinese. Our research demonstrates that Chinese translations with source texts from a typologically distant language (i.e., English) also differ to a large extent from non-translated original Chinese and such differences can be identified by machine learning algorithms.

As to the second research question, we have shown that SVMs is the best-performing classifier in the predicting task, followed by LDA and MLP. RF is the least performing in terms of accuracy. Our results that SVMs is the best-performing classifier corroborated the findings of previous studies [35, 48]. A high success rate (AUC: 90.5% and Accuracy: 84.3%) was achieved with SVMs using entropy-based features. As the two corpora contain four different genres with a total of 500 files respectively, the high accuracy rate achieved in our machine learning models shows that translational language is categorically different from original language irrespective of genre influences. By comparing the performance of a number of classifiers, we are also able to show that different classifiers perform unequally in differentiating translation from non-translation.

## Conclusion

This study was aimed at identifying whether translated Chinese can be distinguished from non-translated original Chinese using entropy measures. By comparing seven types of entropy values in four major genres between translated and non-translated Chinese texts, our study has revealed that translation as a mediated language is categorically different from non-translation in the Chinese language, thus confirming previous hypotheses that translation is unique in its own way [6, 7]. Our study has provided support to this line of inquiry from a Chinese perspective, which is a language relatively underexplored in the existing research. Our study has shown that informational-theoretic indicators such as entropy can be successfully applied to the classification between translational and non-translational languages. Besides, the use of advanced methods such as machine learning techniques have also proved their strengths for the classification tasks than traditional statistical measures.

Although we have added new empirical evidence and understanding to the translationese issue, the research findings are limited to translated Chinese and the use of particular entropy

measures. Like similar caveats explicitly stated in previous research [43], the use of comparable corpora might not be an ideal form for investigating the translationese phenomenon as the findings can be difficult to relate to the different translation universals. For example, a lower wordform entropy can either be attributed to a lower lexical richness inherent in the target language or to the interference from the source language. For this reason, future research can explore the use of other corpora of different languages or operationalize other indicators to enable an in-depth research. Similarly, future research can also be conducted by testing the performance of other machine learning algorithms.

## Author Contributions

**Conceptualization:** Kanglong Liu.

**Data curation:** Rongguang Ye, Rongye Ye.

**Formal analysis:** Rongguang Ye, Liu Zhongzhu, Rongye Ye.

**Investigation:** Rongye Ye.

**Methodology:** Kanglong Liu, Liu Zhongzhu.

**Project administration:** Kanglong Liu, Liu Zhongzhu.

**Software:** Rongguang Ye, Rongye Ye.

**Supervision:** Kanglong Liu.

**Visualization:** Rongguang Ye.

**Writing – original draft:** Kanglong Liu.

**Writing – review & editing:** Kanglong Liu.

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
