## [Decision Letter · Decision Letter 0]

18 Jan 2022

PONE-D-21-26016Entropy-based Discrimination between Translated Chinese and Original Chinese Using Data Mining TechniquesPLOS ONE

Dear Dr. Zhongzhu,

Thank you for submitting your manuscript to PLOS ONE. After careful consideration, we feel that it has merit but does not fully meet PLOS ONE’s publication criteria as it currently stands. Therefore, we invite you to submit a revised version of the manuscript that addresses the points raised during the review process.

We look forward to receiving your revised manuscript.

Kind regards,

Wajid Mumtaz

Academic Editor

PLOS ONE

https://journals.plos.org/plosone/s/file?id=ba62/PLOSOne_formatting_sample_title_authors_affiliations.pdf”

Reviewers' comments:

Reviewer's Responses to Questions

**Comments to the Author**

1. Is the manuscript technically sound, and do the data support the conclusions?

Reviewer #1: Yes

2. Has the statistical analysis been performed appropriately and rigorously? 

Reviewer #1: Yes

3. Have the authors made all data underlying the findings in their manuscript fully available?

Reviewer #1: Yes

4. Is the manuscript presented in an intelligible fashion and written in standard English?

Reviewer #1: Yes

5. Review Comments to the Author

Reviewer #1: Review PONE-D-21-26016

General Comments: In the present study, Liu et al. tried to discriminate between translated- and non-translated Chinese by using the data-driven machines learning methods, in which they extracted seven entropy-based indicators including character, wordform unigram, wordform bigram, wordform trigram, POS (Part-of-speech) unigram, POS bigram, and POS trigram entropy from two balanced Chinese comparable corpora. In fact, they were separating the translated Chinese from non-translated Chinese by adopting the Support Vector Machines (SVMs), Linear discriminant analysis (LDA), Random Forest (RF), and Multilayer Perceptron (MLP) algorithms. The results showed that SVMs is the most robust and effective classifier. Therefore, I think the method is reliable and the results are novel enough. In addition, the Discuss section is substantial and explains the results well. I think the authors are asking an important question that has not yet been thoroughly addressed in the literature, and I would like to further raise some minor points that deserve clarification at this stage.

1. In the Introduction section, although the literary beginning is good, I consider it may not be appropriate in the academic journal (lines 2-6).

2. The authors’ statement that “Xiao and Yue’s study [28] found that translated Chinese fiction contains a significantly greater mean sentence length than native Chinese fiction, which contradicts previous assumption proposed by Malmkjær [29]”, lines 111-113. The assumption need be showed here.

4. The Discussion and Conclusion section should be separated.

In sum, I think the present paper may make a significant contribution to the literature in the field of translation studies. I suggest a minor revision at this stage.

6. PLOS authors have the option to publish the peer review history of their article (what does this mean?). If published, this will include your full peer review and any attached files.

Reviewer #1: No

---

## [Author Response · Author response to Decision Letter 0]

28 Jan 2022

Please see attached file "Response to Reviewers".

---

## [Decision Letter · Decision Letter 1]

7 Mar 2022

Entropy-based Discrimination between Translated Chinese and Original Chinese Using Data Mining Techniques

PONE-D-21-26016R1

Dear Dr. Zhongzhu,

We’re pleased to inform you that your manuscript has been judged scientifically suitable for publication and will be formally accepted for publication once it meets all outstanding technical requirements.

Kind regards,

Wajid Mumtaz

Academic Editor

PLOS ONE

Additional Editor Comments (optional):

Reviewers' comments:

Reviewer's Responses to Questions

**Comments to the Author**

1. If the authors have adequately addressed your comments raised in a previous round of review and you feel that this manuscript is now acceptable for publication, you may indicate that here to bypass the “Comments to the Author” section, enter your conflict of interest statement in the “Confidential to Editor” section, and submit your "Accept" recommendation.

Reviewer #1: All comments have been addressed

2. Is the manuscript technically sound, and do the data support the conclusions?

Reviewer #1: Yes

3. Has the statistical analysis been performed appropriately and rigorously? 

Reviewer #1: Yes

4. Have the authors made all data underlying the findings in their manuscript fully available?

Reviewer #1: Yes

5. Is the manuscript presented in an intelligible fashion and written in standard English?

Reviewer #1: Yes

6. Review Comments to the Author

Reviewer #1: The authors have addressed my all concerns, and all the revised contents were very good. I suggest to accept the present manuscript.

7. PLOS authors have the option to publish the peer review history of their article (what does this mean?). If published, this will include your full peer review and any attached files.

Reviewer #1: No

---

## [Editor Report · Acceptance letter]

12 Mar 2022

PONE-D-21-26016R1 

Entropy-based Discrimination between Translated Chinese and Original Chinese Using Data Mining Techniques 

Dear Dr. zhongzhu:

I'm pleased to inform you that your manuscript has been deemed suitable for publication in PLOS ONE. Congratulations! Your manuscript is now with our production department. 

Kind regards, 

on behalf of

Dr. Wajid Mumtaz 

Academic Editor

PLOS ONE